# Neurogenic Fever after Subarachnoid Hemorrhage in Animal Models: A Systematic Review

**DOI:** 10.3390/ijms241411514

**Published:** 2023-07-15

**Authors:** Ernesto Migliorino, Francesco Nonino, Roberto Amici, Domenico Tupone, Raffaele Aspide

**Affiliations:** 1IRCCS Istituto delle Scienze Neurologiche di Bologna, Anesthesia and Neurointensive Care Unit, 40139 Bologna, Italy; r.aspide@isnb.it; 2IRCCS Istituto delle Scienze Neurologiche di Bologna, Epidemiology and Biostatistic Unit, 40139 Bologna, Italy; 3Department of Biomedical and Neuromotor Science, University of Bologna, 40126 Bologna, Italy; 4Department of Neurological Surgery, Oregon Health and Science University, Portland, OR 97239-3098, USA

**Keywords:** aneurysmal subarachnoid hemorrhage, neurogenic fever, animal disease model, brown adipose tissue, autonomic nervous system

## Abstract

The observation of neurogenic fever resulting from subarachnoid hemorrhage (SAH) in animal models is a useful tool for the interpretation of its pathophysiology in humans, which is still a major challenge in the management of neurocritical patients. This systematic review aims to identify the prognostic factors and pathophysiological elements that determine the onset of neurogenic fever and its severity in animal models. In addition, our study aims to analyze which pharmacological treatments are most effective. All the articles available in Pubmed, Embase, and the Biological Science Collection until August 2021 concerning in vivo experimental studies on SAH animal models, including full texts and abstracts written in English and Italian, were considered. The risk of bias was assessed with SYRCLE’s Risk of Bias tool. In total, 81 records were retrieved; after excluding duplicates, 76 records were potentially relevant. A total of 64 articles was excluded after title and abstract screening. The remaining 12 studies were evaluated as full texts, and 6 other studies were excluded (SAH-induced animal studies without a body temperature assessment). In one study, body temperature was measured after SAH induction, but the authors did not report temperature recording. Therefore, only five studies met the search criteria. The high methodological heterogeneity (different animal species, different temperature measurement methods, and different methods of the induction of bleeding) prevented meta-analysis. Synthesis methodology without meta-analysis (SWiM) was used for data analysis. The total number of animals used as controls was 87 (23 rabbits, 32 mice, and 32 rats), while there were 130 animals used as interventions (54 rabbits, 44 mice, and 32 rats). The presence of blood in the subarachnoid space, particularly red blood cells, is responsible for neurogenic fever; the role of hemoglobin is unclear. The mechanism is apparently not mediated by prostaglandins. The autonomic nervous system innervating brown adipose tissue is undoubtedly implicated in the onset of neurogenic fever. The activation of the central adenosine-1 receptor is effective in controlling the temperature of animals with neurogenic fever (by inhibiting thermogenesis of brown adipose tissue).

## 1. Introduction

Subarachnoid hemorrhage (SAH) is one of the most important causes of brain injury that can lead to serious neurological disability; cognitive impairment; or, in the worst cases, death. It represents 5% of all cases of stroke, and a common cause is the rupture of an intracranial aneurysm, accounting for 80% of all cases of non-traumatic SAH [1]. Other causes of SAH are related to traumatic brain injuries, arteriovenous malformations/fistulae, intracranial arterial dissections, amyloid angiopathy, bleeding diathesis, and drug abuse [2].

Subarachnoid (SA) bleeding, following aneurysmal rupture, may lead to complex and multi-organ dysregulation. Pathophysiological alterations which immediately follow SAH are included in the so-called early brain injury (EBI) [3], encompassing several mechanisms occurring in the first 72 h triggered by the rise in intracranial pressure following SAH and the consequent decrease in cerebral blood flow and in cerebral perfusion pressure [4]. The main features of an EBI, potentially leading to relevant consequences on secondary brain injuries and on the neurologic outcome of patients, are global cerebral ischemia, neuroinflammation, neuronal apoptosis/necrosis, blood–brain barrier disruption, and cerebral edema [5]. Other pathophysiological complications possibly occurring in the acute phase of SAH include rebleeding, hydrocephalus [5], seizure [6], and cardiopulmonary or thermoregulatory dysfunctions [7]; while rebleeding and hydrocephalus significantly worsen the clinical outcome, the presence of seizures at the onset of SAH has not been identified as an early indicator of poor outcome [8]. Thermoregulatory alterations can result in an early rise in body temperature (T), called neurogenic fever (NF) [9].

Up to 70% of patients with SAH develop fever [10], and only in up to 50% of them can an infectious etiology be found [11]. Fever usually occurs early in intensive care unit patients with SAH, mostly in the first 72 h [12], and it is associated with an unfavorable outcome, an increased risk of vasospasm, neurologic disability, cognitive impairment, and increased mortality [10]. The available evidence suggests a correlation between the presence of blood inside the cerebral ventricles (particularly the third ventricle) and the occurrence of fever after SAH [10,12]. Neurogenic fever is less responsive to the pharmacological treatments commonly used for infectious fever, i.e., cyclooxygenase (COX) inhibitors such as paracetamol and non-steroidal anti-inflammatory drugs (for example diclofenac or metamizol) [13,14]. Currently, cooling methods (the rapid infusion of cold fluids, ice, and cold packs) and cooling devices (water or air circulating blankets and intravascular heat exchange systems) are the most common treatment for NF following SAH [13,14,15]. Little is known about the pathophysiology of fever in SAH. In recent years, an increased interest in the pathophysiology of NF has fostered animal research on this topic. The aim of this systematic review (SR) is to assess all the published research in which SA bleeding has been induced in an animal model and in which body temperature has been monitored to clarify the dysregulation leading to NF in SAH and to identify potential therapeutic options.

## 2. Methods

The protocol of this SR was previously published (https://doi.org/10.17605/OSF.IO/X694B (accessed on 7 September 2021)).

Authors defined the following research questions: (i) In SAH animal models, which prognostic factors and pathophysiological elements determine the occurrence of NF and its severity? (ii) In SAH animal models with subsequent fever, which (if any) pharmacological treatment reduces body temperature?

A specific research strategy (Appendix A), developed by an information specialist, was launched on PubMed, Embase, and Biological Science Collection databases from inception to 21 August 2021 to retrieve experimental in vivo studies, including full texts and abstracts published in English or Italian language investigating inducted SA bleeding in an animal model.

To answer the first research question, authors selected studies according to the following inclusion criteria: any experimental design, any kind of animal model, assessment of body temperature before and after hemorrhage induction, and adoption of the occurrence of post-SAH NF and/or neurological impairment and/or mortality as primary outcomes.

To answer the second question, authors selected studies according to the following inclusion criteria: any experimental design and any kind of animal model with NF following SAH in which any pharmacological or non-pharmacological treatment aimed at reducing hyperthermia was used (compared with animal models presenting NF following SAH in which no intervention or an alternative intervention was able to reduce body temperature). The primary outcome was the resolution of fever.

Exclusion criteria were human studies, SAH induction experiments in which quantitative data on body temperature were missing, systematic and narrative reviews, and articles in languages other than English or Italian.

The initial screening, based on title and abstract, was conducted independently by two researchers (ME and AR). Disagreements were resolved by discussion and, if needed, also by a third researcher (NF). Full text of potentially eligible studies was subsequently assessed.

Data extracted were as follows (see Table 1): study characteristics and references, species and number of the animals used, basal values of body temperature (before SAH induction), number of individuals, type of treatments, temperature measurements after treatments and outcome, site of the temperature measurement, time of fever onset after SAH, and duration of fever both in control and in intervention groups. Risk of bias and internal validity of the studies were assessed by two reviewers independently by means of SYRCLE’s Risk of Bias tool [16] (Table 2).

The review was performed in accordance with the Preferred Reporting Items for Systematic Reviews and Meta-Analyses (PRISMA) guidelines [17,18,19,20].

**Table 1 ijms-24-11514-t001:** List of selected manuscripts.

Progressive Nr	Reference	Animal/Strain	Number of Animals	Basal Values	Controls (Number)	Controls (Treatment)	Controls (Measurements)	Interventions (Number)	Interventions (Treatment)	Interventions (Measurement)	Interventions (Outcome)	Site of T Measurement	Time of Fever Onset	Duration of Fever
**1**	(Frosini et al., 1999) [21]	Adult male New Zealand albino rabbits	27	Temperature: 39.01 ± 0.06°C	9	I.c.v. injection of 10 µL of water	ΔT: not significant	9	I.c.v. injection of 10 µg of Hb solution	ΔT: not significant	Not specified	Rectus	Peak at 200 min	24 h
								9	I.c.v. injection of 100 µg of Hb solution	ΔT: 1.05 ± 0.27°C	Not specified			
**2**	(Kotan et al., 2014) [22]	Adult male New Zealand rabbits	18		4	Only general anesthesia	Temperature: 35.4 ± 1.2°C	5	Injection of autologous whole blood into the cisterna magna	Temperature: 39.8 ± 0.6°C	Dead between 5th and 8th days	Not specified	Not specified	Not specified
								9		Temperature: 38.2 ± 0.45°C	Survived until euthanized on 10th day			
**3**	(Aydin et al., 2019) [23]	Rabbits	32		5	Only general anesthesia	T: 35.46°C	7	Injection of 1 mL of autologous blood into the cisterna magna	T: 37.12°C	Euthanized after 7 days from SAH	Brain	Not specified	Not specified
					5	Injection of 1 mL of saline into the cisterna magna	T: 35.65°C	15		T: 38.94°C	Euthanized after 21 days from SAH			
**4**	(Thomas et al., 2020) [24]	Mice			17 wild types	Subarachnoid injection of 60 µL of saline	T_MAX_: 36.8°C	14 wild types	Subarachnoid injection of 60 µL of blood	T_MAX_: 37.5°C FB significantly hyperthermic	Not specified	Peritoneum	Not specified	Not specified
					3 global deletions TLR4		FB not significant	6 global deletions TLR4		FB significantly hypothermic				
					3 myeloid deletions TLR4		FB not significant	6 myeloid deletions TLR4		FB significantly hypothermic				
					3 microglial deletions TLR4		FB not significant	6 microglial deletions TLR4		FB significantly hypothermic				
					6 wild types	I.c.v. injection of LPS	FB significantly hyperthermic	6 neuronal/astrocyte deletions EP3		FB significantly hyperthermic				
								6 wild types + 100 µg of meloxicam		FB significantly hyperthermic				
**5**	(Tupone and Cetas, 2021) [25]	Male Wistar rats		T_CORE_: 37.1 ± 0.2°C	7	Injection of aCSF in SA space	ΔT_CORE_ = not significant T_MAX_ = not significantly different	7	Injection of blood in SA space	ΔT_CORE_ = 1.36 ± 0.1°C T_MAX_ = 38.5 ± 0.2°C	Euthanized after 2 h from SAH	Interscapular area, beneath BAT	Peak at 120 min	Not specified
					5	Sham perforation of ICA	T_MAX_ = not significantly different	5	Perforation of ICA	T_MAX_ = 38.9 ± 0.2°C				
				BAT-SNA: 55.2 ± 22.2% T_BAT_ baseline: 34.6 ± 0.4°C	4	Insertion of needle in SA space	BAT-SNA: 61.8 ± 25.4% ΔT_BAT_ = −0.05 ± 0.06°C	5	Injection of blood in SA space in rats pretreated with indomethacin (iv)	BAT-SNA: 239.3 ± 56.2% ΔT_BAT_ = +0.6 ± 0.2°C				
					6	Injection of aCSF in SA space in rats pretreated with indomethacin (iv)	BAT-SNA: 13.2 ± 8.9% ΔT_BAT_ = +0.1 ± 0.04°C	5	Injection of blood in SA space in rats pretreated with saline (iv)	BAT-SNA: 275.2 ± 106.0% ΔT_BAT_ = +0.9 ± 0.3°C				
				BAT-SNA: 94.6 ± 57.3% T_BAT_ baseline: 35.0 ± 0.7°C	5	Injection of plasma in SA space	BAT-SNA: 40.4 ± 53.5% ΔT_BAT_ = +0.05 ± 0.1°C	4	Injection of RBC in SA space	BAT-SNA: 539.6 ± 192.1% ΔT_BAT_ = +0.7 ± 0.2°C				
					5	Insertion of needle in SA space	BAT-SNA: −1772 ± 2.0% ΔT_BAT_ = +0.08 ± 0.03°C	6	I.C.V. injection of CHA post-SA space injection (blood and RBC)	BAT-SNA: from 349 ± 109.9% to 13.8 ± 10.64% ΔT_BAT_ = -0.33 ± 0.09°C				

**Table 2 ijms-24-11514-t002:** SYRCLE’s Risk of Bias tool.

	Was the Allocation Sequence Adequately Generated and Applied?	Were the Groups Similar at Baseline or Were They Adjusted for Confounders in the Analysis?	Was the Allocation to the Different Groups Adequately Concealed During the Experiment?	Were the Animals Randomly Housed During the Experiment?	Were the Caregivers and/or Investigators Blinded from Knowledge of Which Intervention Each Animal Received During the Experiment?	Were Animals Selected at Random for Outcome Assessment?	Was The Outcome Assessor Blinded?	Were Incomplete Outcome Data Adequately Addressed?	Are Reports of the Study Free of Selective Outcome Reporting?	Was the Study Apparently Free of Other Problems That Could Result in High Risk of Bias?
Study 1	yes	no	unclear	unclear	unclear	unclear	no	unclear	no	no
Study 2	no	unclear	no	unclear	unclear	no	no	yes	no	unclear
Study 3	unclear	unclear	no	unclear	unclear	no	unclear	no	unclear	unclear
Study 4	yes	no	unclear	no	unclear	no	unclear	yes	yes	no
Study 5	yes	yes	no	unclear	no	unclear	no	unclear	yes	yes

## 3. Results

Eighty-one records were retrieved (flow chart—Figure 1). After excluding the duplicates, 76 papers were potentially relevant. Sixty-four papers were excluded after title and abstract screening. The 12 remaining studies were evaluated as full texts, and 6 more studies were excluded (SAH-induced animal studies without a body temperature assessment). In one study, body temperature was measured after SAH induction, but the authors did not report the temperature recording. Finally, only five studies met all the inclusion criteria for this review (Table 1). The authors numbered the papers found with progressive numbers from one to five according to the year of publication.

The species of the animal models used in the selected papers were different: three studies [21,22,23] performed experimentation on rabbits, while two studies [24,25] included rodents (mice and rats).

In each paper, animals were divided into controls and interventions. The total number of animals used as controls was 87 (23 rabbits, 32 mice, and 32 rats), while that of the animals used as interventions was 130 (54 rabbits, 44 mice, and 32 rats).

The treatments fulfilled in the controls were variable; in the interventions, the methods used to induce SA bleeding were different: the intracerebroventricular (i.c.v.) injection of hemoglobin (Hb), injection of autologous blood into the SA space, injection of blood or its derivatives (red blood cells (RBC), plasma) into the SA space at a pre-chiasmatic level, and perforation of the anterior cerebral artery (see Table 1). Moreover, in study 4 [24], genetic manipulations were performed on mice to create special knock-out populations. A subgroup of mice was genetically manipulated to induce multiple deletions of the Toll-like receptor 4 (TLR4) gene, and three subfamilies of mice with various kinds of TLR4 deletions were created: a population with the global deletion of TLR4, a population with a TLR4 deletion in all the myeloid line cells, and a population with a specific TLR4 deletion in the microglial cells. Finally, another population of mice with a specific deletion of the E-prostanoid 3 receptor (EP3R) gene (the prostaglandin E2, PGE2, receptor) in the neurons and astrocytes was created.

The rabbits of the intervention groups were treated as follows: 9 rabbits (study 1) underwent an i.c.v. injection of 10 μg of a Hb solution and did not show a statistically significant increase in body T; 9 rabbits (study 1) underwent an i.c.v. injection of 100 μg of a Hb solution and presented a significant increase in rectal temperature; the remaining 36 rabbits (14 from study 2 and 22 from study 3) were subjected to an SA injection of autologous blood (in correspondence with the cisterna magna). Among the 36 rabbits subjected to an SA injection of blood, only 7 (19.4%, from study 3) did not show a significant increase in body temperature (37.12 °C, measured in the brain), while the remaining 29 (80.6%, 14 from study 2 and 15 from study 3) all presented a body temperature > 38 °C after bleeding. In study 2, among the rabbits given a blood injection, five died between the fifth and the eighth day after the hemorrhage, and eight survived until they were euthanized (tenth day). The rabbits that died between the fifth and the eighth day after hemorrhage had a higher body temperature compared to the rabbits that were euthanized.

In study 4, the intervention group was organized as follows: 14 wild-type (WT) mice; 6 WT mice treated with meloxicam, a cyclooxygenase (COX) inhibitor; 6 mice with a neuronal/astrocyte deletion of EP3R; and 18 mice with a genetic deletion of TLR4 (6 with a global deletion, 6 with a myeloid deletion, 6 with a microglial deletion) underwent an SA injection of blood. WT mice, WT mice treated with meloxicam, and mice with a neuronal/astrocyte deletion of EP3R showed a significant fever burden (FB) after SA blood injection; all the mice with a conditional deletion of TLR4 were hypothermic after SA blood injection, i.e., they showed a negative FB. In this study, body temperature was measured by means of the placement of data loggers in the peritoneal cavity that measured temperature every 15 min for a total of 300 measurements (during a period of 75 h). The FB was calculated by subtracting 36.5 °C from each temperature point and summing the values obtained over the 300 measurements.

In study 5, male Wistar rats were used, the temperature was measured in the rectum (core temperature) and directly in brown adipose tissue (T-BAT), and the electrophysiological activity of the sympathetic nerve of the brown adipose tissue (BAT-SNA) was monitored. The rats of the control group underwent the following procedures: (1) an SA injection of artificial cerebrospinal fluid at the level of the pre-chiasmatic cisterns and (2) a sham perforation of the anterior cerebral artery. The rats in the intervention group underwent SAH through two methods: (1) a direct injection of blood (and next, RBC and plasma) into the SA space in correspondence with the basal cisterns at a pre-chiasmatic level and (2) a perforation of the internal carotid artery (ICA). Both rats models (SA blood injection and ICA perforation) showed a significant increase in core temperature compared to the controls. The authors repeated the SA injection of blood in the rats pre-treated with intravenous indomethacin (a COX inhibitor)—these rats showed a significant increase in BAT-SNA and T-BAT—as well as in the rats pre-treated with intravenous saline which underwent SA blood injection; the controls (a simple insertion of a needle in the SA space and a vehicle injection into the SA space of the rats pre-treated with indomethacin) did not show any increase in BAT-SNA and T-BAT. Next, the authors separated the blood components into RBC and plasma: only an RBC injection in the SA space caused a significant increase in BAT-SNA and in T-BAT, while a plasma injection did not show such results. Interestingly, an i.c.v. injection of N6-cyclohexyladenosine (CHA), an agonist of the adenosine A1 receptors (A1AR), caused a decrease in T-BAT and BAT-SNA, previously increased by an RBC SA injection.

One of the main features characterizing the selected studies is heterogeneity: they include different animal species (rabbits, mice, and rats); the kinds of treatments in the controls and interventions are variable; and the modalities, times, and sites of the temperature measurements are different. Therefore, the authors decided not to perform a meta-analysis. For this reason, the compilation method used for this review followed the synthesis without meta-analysis (SWiM) methodology [26] both for the description of the methods and for the analysis of the results.

## 4. Discussion

The analysis of four [22,23,24,25] out of five papers found shows that an injection of blood into the SA space significantly increases the core body temperature compared to a vehicle (i.e., saline solution) injection. One study [25] showed that RBC specifically are responsible for the rise of the body temperature when injected into the SA space, while plasma does not elicit such a response. These are the only pre-clinical studies highlighting the correlation between SA bleeding and fever, while this correlation was known in the human clinical literature: in a study by Fernandez et al. [10], intraventricular hemorrhage following SAH was associated with the occurrence of fever as confirmed by Mees et al. [27].

Only in one study [22] is it possible to find a correlation between fever and mortality: the rabbits that died more rapidly had a higher temperature compared to the rabbits that survived until they were euthanized. This finding agrees with data on the correlation between fever following SAH and increased mortality in human clinical studies [10,28].

In one study [21] the role of Hb was examined. Different amounts of Hb were injected into the SA space of the rabbits, and this caused a significant rise in body temperature. It is known that heme, released from Hb molecules into the SA space after SAH, is metabolized by Heme Oxygenase 1 to iron (Fe^2+^), biliverdin, and carbon monoxide [29] in the SA space. Iron deposits have a role in the onset of an EBI by activating neuronal and glial cell membrane damage via free radicals [30,31], but there is a lack of evidence in the literature on iron as a pathophysiologic element in the occurrence of NF.

Since PGE2 in particular has a pivotal role in the occurrence of septic fever [32], its possible involvement in NF has been considered in two studies [24,25]; these studies show for the first time that NF after SAH occurs in animals pretreated with a COX inhibitor and in animals with a lack of EP3R, demonstrating that NF is not PGE2-mediated as in septic fever [33,34,35].

In study 4 [24], the role of TLR4 was explored. TLR4 is part of the Toll-like receptor family and gives origin to the intracellular signaling cascade that leads to the pro-inflammatory process [36] and to the production of PGE2 following an infection [37]. TLR4 is expressed from the antigen-presenting cells, granulocytes, macrophages, microglia, and also from neurons and astrocytes [38]: when TLR4 links the surface structures of the bacteria (pathogen-associated molecular patterns, represented specifically by the lipopolysaccharides), it induces the activation of the macrophages, with the production of inflammatory cytokines and chemokines, which is responsible for the innate immune response and for the occurrence of septic fever [32,39]. TLR4 also recognizes other molecules, like heme and fibrinogen, which are released during SAH [40], so it represents an important molecular element for neuroinflammation. The authors induced SAH in mice with a conditional deletion of the TLR4 gene in different cell lines, obtaining global hypothermia (a negative FB) and deducing that microglia TLR4 was necessary for the occurrence of NF in SAH. Nevertheless, this is not sufficient to prove the role of TLR4 in the occurrence of fever. The measured core temperature is a resultant deriving from the balance between thermoregulatory effector activity (i.e., skin vasomotion, sweat regulation, BAT thermogenesis, and shivering thermogenesis) and the passive heat exchange between the body and the environment (balance point) [41]; fever is when the balance point is elevated, due to an activation of the thermoeffectors, and the core body temperature overcome 38 °C [42]. A negative FB does not necessarily imply a reduction or an absence of thermoeffector activity following SA bleeding, only a low body temperature (the authors define mice with a negative FB as hypothermic, that is, a body temperature lower than 35 °C [43]). SA blood may induce a thermoregulatory response in hypothermic mice with a TLR4 deletion and cause an increase in the balance point that nevertheless does not overcome the threshold for the fever definition. In this sense, the role of TLR4 in NF occurrence is not completely clear.

The authors of study 5 [25] investigated the role of the autonomic nervous system. The neurophysiology of thermoregulation has been largely described [44], and an essential feature responsible for the febrile response in rodents is BAT thermogenesis: the central neural circuits (mainly located in the preoptic area of the hypothalamus [45]) regulate the activity of the BAT sympathetic nerve, increasing or inhibiting BAT thermogenesis [46]; the presence of blood in the SA space, and more specifically of RBC, is directly responsible for a significant rise in BAT-SNA and subsequently in rats’ body temperature, while a plasma SA injection was not, suggesting that RBC membrane components may someway trigger the central pathways and neurocircuitry of thermoregulation, leading to fever onset.

They experimented with the role of central A1AR in blocking NF. The activation of the A1AR in rats, via the i.c.v. administration of CHA, induces a state of deep hypothermia by inhibiting BAT and shivering thermogenesis [47]. After the onset of NF caused by SAH, the authors realized the i.c.v. injection of CHA, obtaining a reduction in BAT-SNA and in BAT thermogenesis and indicating that the neurophysiologic circuits involved in NF following SAH may be the same as those involved in body temperature regulation.

Some limitations of this systematic review can be identified: limitations related to our research method and limitations related to the quality of the selected studies.

The limitations of our research method include the following: the inclusion criteria on language was restricted to English and Italian; we used only three search engines (PubMed, Embase, and the Biological Science Collection) because of the skills of our information specialist with these three and because we did not consider other search engines to be potentially useful; and our field of research was restricted to studies published by specialized journals, and we did not consider publications by governance agencies, pharmaceutical companies, etc.

Limitations related to the quality of the selected studies deal with the risk of bias in these studies. We used SYRCLE’s Risk of Bias tool for animal studies [16] to evaluate the quality of the studies found, and some critical issues emerged (Table 2): the risk of bias of the selected studies is not clear in all the cases, whereby it is difficult to evaluate the quality of the studies found; the methods of randomization of the animals are not specified in all the studies found, and also baseline characteristics (such as body temperature) are not correctly specified; the methods and sites of temperature recording are heterogeneous, making it difficult to interpret data found; and in all studies, there is no blinding of the investigators, and this involves a risk of detection bias. In some studies (studies 2 [22] and 3 [23]), temperature recording after SAH was not the main objective of the experimentation, so the quality of data reported is inferior; the outcome of animal models after SAH is not correctly described in all the studies both in terms of neurological impairment and in terms of mortality. Improving the quality of the pre-clinical research and standardizing the species of animal models; the baseline characteristics and conditions of the animals used; the methods of SAH induction; the sites, modalities and timing of temperature recording; and the outcome assessment of the animal models may lay the groundwork for a possible translation in clinical pharmacology and human clinical research; in particular, future pre-clinical research should be oriented toward a better comprehension of which components of RBC are responsible for the occurrence of NF after SAH and toward the identification of a possible pharmacological agent aimed at reducing BAT-SNA activation following SAH, thereby blocking the onset of the NF. The role of TLR4 and the mediators of neuroinflammation (heme, iron, and fibrinogen) must be further investigated by seeking their possible physiopathological role in thermoregulatory dysfunction after SAH.

## 5. Conclusions

Evidence from pre-clinical studies on the pathophysiology of NF in SAH is poor. There are only five papers that have studied the occurrence of NF after SAH in the animal model, and there is large methodological inhomogeneity. The presence of blood (RBC in particular) in the SA space is correlated with the occurrence of NF. The neural pathways responsible for NF occurrence seem to be the same as those implicated in septic fever, at least in its efferent component (BAT-SNA). In contrast to septic fever, NF seems to not be mediated by PGE2. The role of TLR4 is unclear and must be further studied in animal models. Central A1AR seems to have a role in blocking NF by means of its agonist CHA; a better understanding of its function in the neural circuit of thermoregulation is needed, and this might open new scenarios for possible future human studies. The studies available present some limitations related to different kinds of bias; improving the quality and standardizing conditions of experimentations may lead to possible human clinical research translation.

## Figures and Tables

**Figure 1 ijms-24-11514-f001:**
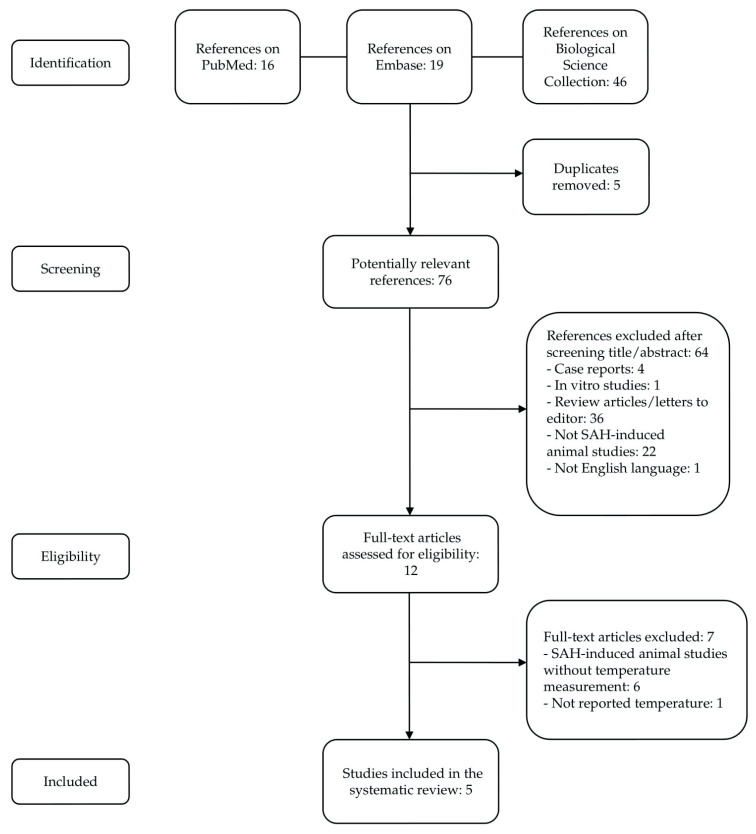
PRISMA flow chart.

## Data Availability

The data supporting this study’s findings are available from the corresponding author, EM, upon reasonable request.

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
