# Peer review of "Neurogenic Fever after Subarachnoid Hemorrhage in Animal Models: A Systematic Review"

_ijms, 2023, doi:10.3390/ijms241411514_

Round 1

Reviewer 1 Report

This paper represents a systematic review on the neurogenic fever after subarachnoid hemorrhage in animal models.

The topic is relevant in the field as it is a clinical problem in humans, experimental animal studies were included. This is a systematic review of a published material.

Tables are typical for systematic review, figure (PRISMA flow-chart) is also correctly done.

Conclusions provides an answer to the main question in details and the references are appropriate.

The systematic review was performed in a standard manner, there are no doubts about methodology.

Author Response

Thank's to the reviewer for the comments on our paper.

We've done the minor revisions requested for our manuscript.

Reviewer 2 Report

The authors present a systematic review with the aim of estimating the prognostic factors and pathophysiological elements that determine the onset of neurogenic fever and its severity after subarachnoid hemorrhage in the animal model. In total, 5 studies met the search criteria of the review, which concluded that the presence of blood (in particular, red blood cells) in the SA space correlates with the occurrence of neurogenic fever (NF) and the autonomic nervous system innervating brown adipose tissue is undoubtedly implicated in the onset of NF. Activation of the central adenosine-1 receptor is effective in controlling temperature in animals with neurogenic fever. The study is potentially  interesting, but can be improved if the following considerations are addressed: 

- Data on the clinical relevance of the presence of early seizures on clinical outcome or in-hospital mortality in subarachnoid hemorrhage in humans  should be included (see data and comment on the study published in Clin Neurol Neurosurg 1999; 101: 100-5). 

-The authors’ opinion on the future direction of research on this topic should be added in the text 

-Please, rewrite the bibliographic references according to the instructions of Int J Mol Sci . 

Author Response

Thank's to the reviewer for the comments and suggestions.

We have added data about the clinical relevance of the early occurrence of seizures on the outcome of the SAH patients (and also the reference suggested) in the section "Introduction" at lines 55-57.

We have added our opinion about the future direction of the research on this topic in the section "Discussion" at lines 324-330.

We have rewritten the bibliographic references according to the instruction of Int J Mol Sci.